# EIGEN MEMORY TREES

## ABSTRACT

This work introduces the Eigen Memory Tree (EMT), a novel online memory model for sequential learning scenarios. EMTs store data at the leaves of a binary tree and route new samples through the structure using the principal components of previous experiences, facilitating efficient (logarithmic) access to relevant memories. We demonstrate that EMT outperforms existing online memory approaches, and provide a hybridized EMT-parametric algorithm that enjoys drastically improved performance over purely parametric methods with nearly no downsides. Our findings are validated using 206 datasets from the OpenML repository in both bounded and infinite memory budget situations.

## 1 INTRODUCTION

A sequential learning framework (also known as online or incremental learning (Hoi et al., 2021; Losing et al., 2018)) considers a setting in which data instances $x_t \in \mathbb{R}^d$ arrive incrementally. After each instance, the agent is required to make a decision from a set of $|\mathcal{A}|$ possibilities, $a_t \in \mathcal{A}$. The agent then receives scalar feedback $y_t$ regarding the quality of the action, and the goal is for the agent to learn a mapping from $x_t$ to $a_t$ that maximizes the sum of all observed $y_t$.

This general paradigm accommodates a wide array of well-studied machine learning scenarios. For example, in online supervised learning, $\mathcal{A}$ is a set of labels—the agent is required to predict a label for each $x_t$, and the feedback $y_t$, indicates the quality of the prediction.

In a contextual bandit or reinforcement learning setting, $x_t$ acts as a context or state, $a_t$ is an action, and $y_t$ corresponds to a reward provided by the environment. Contextual Bandits have proven useful in a wide variety of settings; their properties are extremely well studied (Langford & Zhang, 2007) and have tremendous theoretical and real-world applications (Bouneffouf et al., 2020).

Regardless of the particulars of the learning scenario, a primary consideration is sample complexity. That is, how can we obtain the highest-performing model given a fixed interaction budget? This often arises when agents only receive feedback corresponding to the chosen action $a_t$, i.e. *partial feedback*. Here, after an interaction with the environment, the agent does not get access to what the best action in hindsight would have been. As a consequence, learners in a partial-feedback setting need to explore different actions even for a fixed $x_t$ in order to discover optimal behavior.

Recent work in reinforcement learning has demonstrated that episodic memory mechanisms can facilitate more efficient learning (Lengyel & Dayan, 2007; Blundell et al., 2016; Pritzel et al., 2017; Hansen et al., 2018; Lin et al., 2018; Zhu et al., 2020). Episodic memory (Tulving, 1972) refers to memory of specific past experiences (e.g., what did I have for breakfast yesterday). This is in contrast to semantic memory, which generalizes across many experiences (e.g., what is my favorite meal for breakfast). Semantic memory is functionally closer to parametric approaches to learning, which also rely on generalizations while discarding information about specific events or items.

This paper investigates the use of episodic memory for accelerating learning in sequential problems. We introduce Eigen Memory Trees (EMT), a model that stores past observations in the leaves of a binary tree. Each leaf contains experiences that are somewhat similar to each other, and the EMT is structured such that new samples are routed through the tree based on the statistical properties of previously encountered data. When the EMT is queried with a new observation, this property affords an efficient way to compare it with only the most relevant memories. A learned "scoring" function $w$ is used to identify the most salient memory in the leaf to be used for decision making.

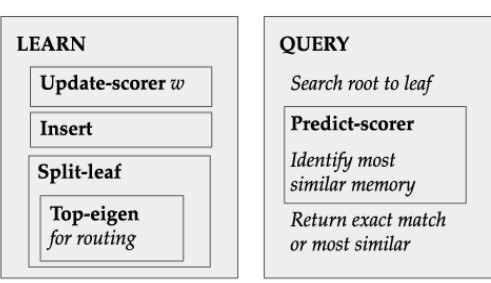

**Figure 1:** A schematic of the Eigen Memory Tree (EMT) algorithm. The outer boxes indicate the two operations supported by EMT, Learn and Query. The inner boxes show the high-level subroutines that occur to accomplish the outer operation.

**Table 1:** Cells indicate the number of datasets in which the row algorithm beats the column algorithm with statistical significance. The stacked algorithm we propose (PEMT) has the most wins in each column as indicated by bold.

|  | PEMT | EMT | PCMT | CMT | Parametric |
|---|---|---|---|---|---|
| PEMT | — | **167** | **87** | **196** | **110** |
| EMT | 31 | — | 42 | 177 | 44 |
| PCMT | 4 | 156 | — | 192 | 75 |
| CMT | 7 | 11 | 10 | — | 17 |
| Parametric | 8 | 151 | 10 | 184 | — |

Specifically, this work

- introduces the Eigen Memory Tree, an efficient tool for storing, accessing, and comparing memories to current observations (see Figure 1).
- shows that the EMT gives drastically improved performance over comparable episodic memory data structures, and sometimes even outperforms parametric approaches that have no explicit memory mechanism.
- proposes a simple combination EMT-parametric (PEMT) approach, which outperforms both purely parametric and EMT methods with nearly no downsides (see Table 1).

In the following section, we introduce the Eigen Memory Tree and overview the algorithms required for storing and retrieving memories. A schematic of EMT and its high-level algorithms can be seen in Figure 1. With this in mind, Section 3 describes related work.

Section 4 follows with an exhaustive set of experiments, demonstrating the superiority of EMT to previous episodic memory models and motivating the EMT-parametric (PEMT) method, a simple and powerful hybrid strategy for obtaining competitive performance on sequential learning problems. Importantly, we show that the PEMT performance advantage holds even when it is constrained to have a fixed limit on the number of memories it is allowed to store.

All experiments here consider the contextual bandit setting, but EMTs are applicable in broader domains as well. This section continually references Table 1, identifying which methods outperform which other methods by a statistically significant amount over all datasets and replicates. We consider a substantial number of datasets from the OpenML (Vanschoren et al., 2014) repository.

Section 5 summarizes our findings, discusses limitations, and overviews directions for future work.

## 2 EIGEN MEMORY TREE

As with episodic memory, EMT is structured around storing and retrieving exact memories. We formalize this notion as the *self-consistency* property: if a memory has been previously INSERTED, then a QUERY with the same key should return the previously inserted value. The self-consistency property encodes the assumption that the optimal memory to return, if possible, is an exact match for the current observation.

EMT is a memory model with four key characteristics: (1) self-consistency, (2) incremental memory growth, (3) incremental query improvement via supervised feedback, and (4) sub-linear computational complexity with respect to the number of memories in the tree. As we discuss in the literature review below, this combination of characteristics separates EMT from previous approaches.

**Memory.** EMT represents memories as a mapping from keys to values, $\mathcal{M} := \mathbb{R}^d \to \mathbb{R}$, where $d$ is the dimensionality of the context $x_t$, and $y_t \in \mathbb{R}$ corresponds to observed feedback. A query to the EMT requires an $x_t$ and returns a previously observed value $\hat{y} \in \mathbb{R}$ from its bank of memories. EMT learning, which updates both the underlying data structure and the scoring mechanism, requires a full $(x_t, y_t)$ observation pair $\mathcal{M}$.

---

**Algorithm 1** Primary Functions

---

1: **class** Node (router $\in \mathbb{R}^d$, boundary $\in \mathbb{R}$, left: Node, right: Node, M $\subseteq \mathbb{R}^d \times \mathbb{R}$):
2: **Initialize:** $root \leftarrow$ Node()
3: **Initialize:** $w \leftarrow$ random vector in $\mathbb{R}^d$
4: **Initialize:** $c \leftarrow$ leaf capacity

5: **function** QUERY($x \in \mathbb{R}^d$)
6:     $n \leftarrow root$
7:     **while** $n$ is not leaf $n \leftarrow n$.left **if** $\langle n.\text{router}, x \rangle \leq n$.boundary **else** $n$.right
8:     **return** $\arg\min_{(x_m, y_m) \in n.\text{M}}$ PREDICTSCORER($x, x_m$)
9: **end function**

10: **function** LEARN($x \in \mathbb{R}^d$, $y \in \mathbb{R}$)
11:     $n \leftarrow root$
12:     **while** $n$ is not leaf $n \leftarrow n$.left **if** $\langle n.\text{router}, x \rangle \leq n$.boundary **else** $n$.right
13:     UPDATESCORER($x$,$y$)
14:     $n.\text{M} \leftarrow n.\text{M} \cup (x, y)$                     ▷ Insert memory into leaf.
15:     **if** $|n.\text{M}| \geq c$ **then**
16:         SPLITLEAF($n$)
17:     **end if**
18: **end function**

---

**Data structure.** The underlying data structure used by the EMT is a binary tree. The tree is composed of a set of nodes, $\mathcal{N}$, each of which is either an internal node or a leaf node. If a node $n \in \mathcal{N}$ is internal it possesses two child nodes ($n.\text{left} \in \mathcal{N}$ and $n.\text{right} \in \mathcal{N}$), a decision boundary $n.\text{boundary} \in \mathbb{R}$, and a top eigenvector $n.\text{router} \in \mathbb{R}^d$. The decision boundary and top eigenvector, as discussed later, are used to route queries through the tree. If a node $n$ is a leaf node then it will instead possess a finite set of memories $n.\text{M} = \{(x, y) \in \mathbb{R}^d \to \mathbb{R}\}$. In addition to the nodes and routers EMT also possesses a single global scoring function responsible for selecting which memory to return given a query and leaf node. This function is parameterised in by the weight vector $w \in \mathbb{R}^d$.

**Learn.** Newly acquired information is stored during the learning operation (Line 10 in Algorithm 1), which takes an item $\mathcal{M}$ and traverses the tree until a suitable insertion leaf is identified (Line 14 in Algorithm 1). Immediately before insertion, the scoring function weights $w$ are updated such that the scores of the insertion leaf's memories improve with respect to the observed $x_t$ and $y_t$ (Line 13 in Algorithm 1)

**Scorer.** EMT's scorer, $w$, ranks candidate memories at query time. The scorer supports both predictions (see Line 12 Algorithm 2) and updates (see Line 1 Algorithm 2). Intuitively, the scorer can be thought of as a dissimilarity metric, assigning small values to similar query pairs and large values to dissimilar query pairs.

A pair of identical query vectors is guaranteed to result in a score of $0$, satisfying the self-consistency property mentioned earlier. This is achieved for any two $x_1$ and $x_2$ by applying the scorer on the coordinate-wise absolute value difference between pairs, $z \leftarrow |(x_1)_i - (x_2)_i|\}_{i=1}^d$, which is all zeros if the two contexts are identical. Predictions are then made by linearly regressing this vector onto the scorer's weights, and clipping predictions $\langle w, z \rangle$ such that the minimum is $0$. The clipping operation ensures that predictions will be $0$ even when weights are negative.

Updating the scoring function is done via a *ranking loss*, which considers (1) $(x_{\text{best}}, y_{\text{best}})$, which is the memory-reward pair that *would* be retrieved by the scorer currently and (2) $(x_{\text{alt}}, y_{\text{alt}})$, an alternative memory in the same leaf that has a reward most similar to the current observation (omitting the retrieved memory $x_{\text{best}}$). If $y_{\text{alt}}$ gives a better prediction of the observed reward than $y_{\text{best}}$, we adjust the scorer to decrease the learned dissimilarity between $x_{\text{alt}}$ and the current target $x_t$. Alternatively, if the retrieved memory $y_{\text{best}}$ is closer to the observed reward, we do the opposite, emphasizing the similarity between $x_t$ and $x_{\text{best}}$ and making $x_t$ and $x_{\text{alt}}$ more dissimilar. Specifically, this is done by taking a gradient step with respect to an $L_2$ loss that encourages more similar pairs to have a score near $0$ and more dissimilar pairs to have a score near $1$.

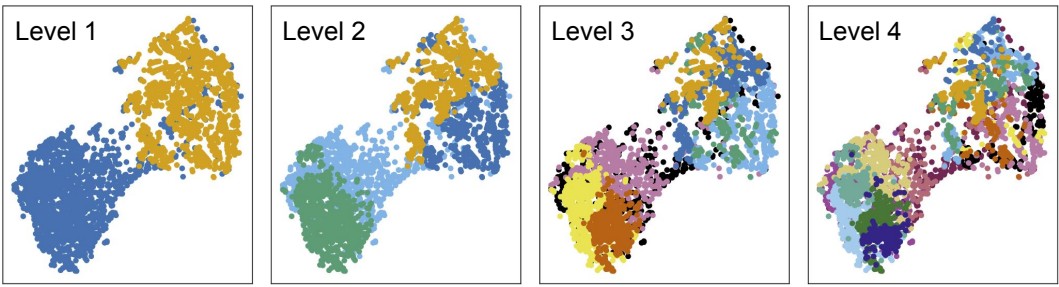

**Figure 2:** A t-SNE visualization of an EMT after training on OpenML's BNG (breast-w) dataset. Each point represents a memory and is colored to indicate the node at which the memory is stored. Each plot shows memories colored by how they are allocated at the indicated level of the tree. The level of the tree considered increases by one from left to right, doubling the number of nodes considered (number of data paths from root to node).

---

**Algorithm 2** Scorer Functions

---

**function** UPDATESCORER($M \subseteq \mathbb{R}^d \times \mathbb{R}, x \in \mathbb{R}^d, y \in R$)
    $(x_b, y_b) = \arg\min_{(x_m, y_m) \in M} \text{PREDICTSCORER}(x, x_m)$
    $(x_a, y_a) = \arg\min_{(x_m, y_m) \in (M \setminus (x_b, y_b))} |y - y_m|$
    **if** $|y_b - y_t| < |y_a - y_t|$ **then**
        $\mathcal{L} = \text{PREDICTSCORER}(x_b, x)^2 + (1 - \text{PREDICTSCORER}(x_a, x))^2$
        $w \leftarrow w - \eta \frac{\partial \mathcal{L}}{\partial w}$
    **else if** $|y_b - y_t| > |y_a - y_t|$ **then**
        $\mathcal{L} = \text{PREDICTSCORER}(x_a, x)^2 + (1 - \text{PREDICTSCORER}(x_b, x))^2$
        $w \leftarrow w - \eta \frac{\partial \mathcal{L}}{\partial w}$
    **end if**
**end function**

**function** PREDICTSCORER($x_1 \in \mathbb{R}^d, x_2 \in \mathbb{R}^d$)
    $z \leftarrow \{|(x_1)_i - (x_2)_i|\}_{i=1}^d$                          $\triangleright z \in \mathbb{R}^d$
    **return** $\max(0, \langle w, z \rangle)$                $\triangleright$ We clip for self-consistency purposes.
**end function**

---

**Query.** To predict $y_t$ we can call EMT's QUERY operation with $x_t$ (see Line 5 in Algorithm 1). Queries to the EMT trigger a search over the internal data structure, beginning at the root node and routing through internal nodes until a leaf is eventually reached. When the search procedure is presented with an internal node $n$, routing follows a simple rule: if $\langle n.\text{router}, x_t \rangle \leq n.\text{boundary}$ we proceed left, if not we proceed right (see Line 7 in Algorithm 1). When we arrive at a leaf the scorer is then called to identify the most similar memory available (see Line 8 in Algorithm 1).

**Eigen-router initialization.** An important hyperparameter in the EMT is the maximum leaf capacity $c$, which controls the number of memories stored in a single leaf node. Once the leaf reaches this capacity, the corresponding node is assigned a left and right child, and its memories are allocated across them. If $c$ is small, the tree is extended frequently to accommodate new data, and when it is large, the tree grows more slowly. A large $c$ allows many memories to be scored for a particular query, which can improve statistical performance but may damage efficiency. Correspondingly, smaller capacities permit fewer memories to be scored for a given query, which may improve computational efficiency by sacrificing predictive performance.

We check whether a leaf node $n$ is at capacity each time a memory is added to it (i.e., $|n.\text{M}| \geq c$). If it is, the leaf is split, turning $n$ into an internal node. The splitting process begins by assigning routing behavior to the node, which governs how new samples will traverse the tree. This is done by (1) approximating the first principal component of its stored memories and (2) computing the median value of memories projected onto this vector. When a new sample arrives, we project it onto this eigenvector and route the sample left if the corresponding value is less than or equal to this median and otherwise route it right (see Line 3 in Algorithm 3).

We follow the same rule for distributing memories across the children of a newly split node—memories with a projection less than or equal to this median are stored in the left child

---

**Algorithm 3** Leaf Functions

---

**function** SPLITLEAF($n$: Node)
    $n$.router $\leftarrow$ TOPEIGEN($n$.M)
    $n$.boundary $\leftarrow$ median($\{\langle n.\text{router}, x_m \rangle | (x_m, y_m) \in n.\text{M}\}$)
    $n$.left.M $\leftarrow \{(x_m, y_m) \in n.\text{M} \mid \langle n.\text{router}, x_m \rangle \leq n.\text{boundary}\}$
    $n$.right.M $\leftarrow \{(x_m, y_m) \in n.\text{M} \mid \langle n.\text{router}, x_m \rangle > n.\text{boundary}\}$
    $n$.M $\leftarrow \emptyset$
**end function**

**function** TOPEIGEN($M \subseteq \mathbb{R}^d \times \mathbb{R}$)
    $X = \{x_m \mid (x_m, y_m) \in M\}$                                      $\triangleright X \in \mathbb{R}^{d \times |M|}$
    $X = X(I_{|M|} - \frac{1}{|M|}\mathbf{1}_{|M|})$                       $\triangleright$ Mean center the rows of $X$
    $v \leftarrow$ random unit vector in $\mathbb{R}^{|M|}$
    **for** $n \leftarrow 1$ to $|M|$ **do**
        $v \leftarrow \frac{v + \frac{1}{n}(X_{:,n})(X_{:,n})^\top v}{||v + \frac{1}{n}(X_{:,n})(X_{:,n})^\top v||}$
    **end for**
    **return** v
**end function**

---

and others are stored in the right. The approximate eigenvector assigned to $n$.router is computed using Oja's method, a popular approach for performing PCA on streaming data (Balsubramani et al., 2013) as shown in Algorithm 3. Because the decision boundary $n$.boundary is the median of the projected memories, the allocation of data across children in a newly split node is nearly even; hopefully resulting in a well-balanced tree. A plot of $4,000$ memories within a tree can be seen in Figure 2.

## 3 RELATED WORK

This section overviews alternative memory structures through the lens of our desiderata: self-consistency, incremental growth, learning, and sub-linear computational complexity. Two illuminating extreme points are associative data structures (e.g., hashmaps) that do not learn and only support exact retrieval; and supervised models (e.g., ordinary least squares) which compile experience into a structure which supports fast retrieval, but cannot guarantee self-consistency.

Classic nearest-neighbor models (Friedman et al., 1975) are self-consistent and grow incrementally, but do not learn (the metric) and have poor computational complexity; considerable attention has unsurprisingly been put towards improving nearest neighbors algorithms along these axes. Exact or approximate nearest-neighbor methods, e.g., Beygelzimer et al. (2006); Ram & Sinha (2019); Datar et al. (2004); Dasgupta & Sinha (2013), can reduce computational complexity but (beyond inserting new memories) do not learn. Weinberger et al. (2005) learn a metric for use with nearest-neighbors but does not learn incrementally. Analogously, learned hash functions (Salakhutdinov & Hinton, 2009; Rastegari et al., 2012) produce an associative data structure but learning is not incremental.

Differentiable neural memory systems learn an associative mapping using gradient-based optimization (Sukhbaatar et al., 2015; Graves et al., 2016). Unfortunately their computational demands are severe, preventing practical applications with large memories and prompting variants that use custom hardware (Ni et al., 2019; Ranjan et al., 2019) or exploit sparsity (Karunaratne et al., 2021).

The only memory model in the literature possessing all of EMT's desiderata is the contextual memory tree (CMT) (Sun et al., 2019). EMT and CMT both use an internal tree structure with routers and a global scorer. EMT differs from CMT in three ways (1) EMT uses fixed top-egeinvector routers while CMT learns routers incrementally, (2) the EMT scorer uses pairwise feature differences rather than CMT's feature interactions (the importance of this can be seen in our own ablation studies below) and (3) the EMT scorer is updated with respect to a ranking loss function rather than CMT's squared loss. As discussed later, (1) and (2) together ensure a strong self-consistency constraint, such that exact memories are returned if available. Because CMT updates routing mechanisms, even exact matches for a particular observation may be inaccessible.

Finally, it remains an open question how to best use episodic memory. For example, the work of (Lengyel & Dayan, 2007) evaluates directly using memories to decide how to act in a sequential

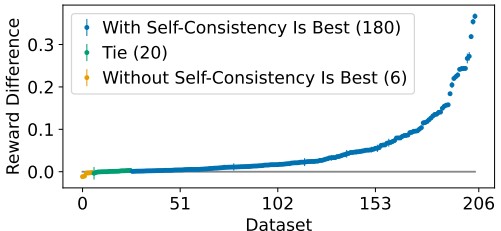 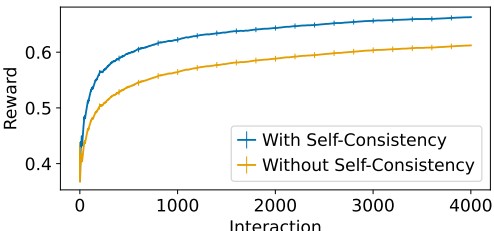

**Figure 3:** A per-dataset comparison of final progressive reward between EMT-CB and a variant of EMT-CB without self-consistency. Each point is a single dataset. The y-axis shows the difference between the two learners. The x-axis is ordered by this difference.

**Figure 4:** Progressive reward for EMT-CB and a variant of EMT-CB without self-consistency. Each line is an average over 206 datasets in the OpenML repository. Progressive reward is shown as a function of interactions and vertical lines are standard error.

environment. Other work has used episodic memory as a tool to support a supervised learner either as input into a non-parametric model (Blundell et al., 2016; Pritzel et al., 2017) or as a tool to train a supervised learner (Lin et al., 2018). Still more has treated episodic memory as a complement to supervised learning and the goal is to combine the two model's predictions Hansen et al. (2018); Feng et al. (2017). Our work evaluates EMT in terms of its performance as a direct predictor (EMT-CB) and as a complementary predictor to a parametric model (PEMT-CB).

## 4 EXPERIMENTS

The experiments in this section consider the Contextual Bandit (CB) framework, a setting in which an agent is sequentially presented with a $d$-dimensional "context" vector $x_t$ and is required to execute one of $|\mathcal{A}|$ actions. For each executed action $a_t \in \mathcal{A}$, the agent receives a corresponding reward $y_t$. The goal of the agent is to maximize the total reward received over the course of an evaluation.

The EMT algorithm presented above is a general-purpose supervised learning algorithm, and not an algorithm for solving CB problems specifically. We however can easily adapt EMT for the CB setting by augmenting it with an exploration algorithm (see Algorithm 4). For our experiments we used an $\epsilon$-greedy exploration algorithm with $\epsilon = .1$. That is, on each iteration with probability $\epsilon$ an action in $\mathcal{A}$ was selected with uniform random probability. Otherwise for each $a \in \mathcal{A}$ EMT is queried and the action with the highest predicted reward is taken. We will call this adaptation EMT-CB. Code to reproduce all results is available at `redacted`.

**Datasets** We consider 206 contextual bandit problems derived from OpenML (Vanschoren et al., 2014) classification datasets via a supervised-to-bandit transformation (Bietti et al., 2021). OpenML data are released under a CC-BY[1] license. For large datasets, we consider a random subset of $4,000$ training examples. We also scale the $i$-th feature of every sample $x_t^i$ in each dataset by $1/(\max_t x_t^i - \min_t x_t^i)$. For each problem, the action set $\mathcal{A}$ is is the set of classification labels. When an agent correctly classifies an example a reward of one is given otherwise a reward of zero is given.

**Evaluation** Learners are evaluated online via progressive loss (Blum et al., 1999). Specifically, each time the agent receives reward, we update the total reward earned, normalized by the total number of interactions. After $T$ total interactions, the progressive reward is $1/T \sum_{t=0}^{T} y_t$. These progressive rewards are plotted throughout the course of this section. Additionally, in order to calculate confidence bounds every data set is evaluated 50 times with a different random seed for each.

### 4.1 SELF-CONSISTENCY

The EMT incorporates two particular design decisions: (1) using eigenvector routing and (2) using a scorer where self-consistency is a hard constraint, which as mentioned earlier, guarantees that an exact-match memory is always retrieved as long as one is available.

To examine the importance of self-consistency, we compare EMT to a variant that does not include this property. In this variant, we modify Line 13 of Algorithm 2 such that the input to the regressor,

---

[1] `https://creativecommons.org/licenses/by/2.0/`

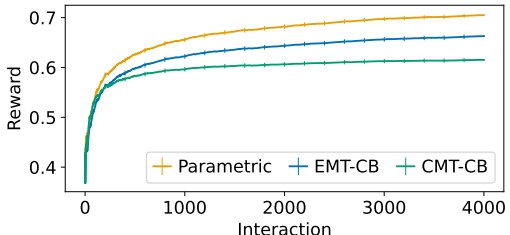
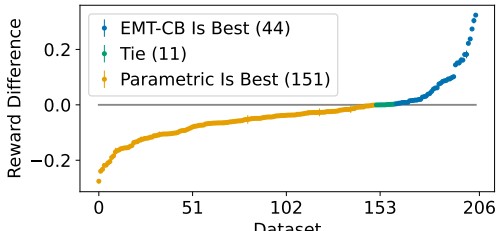

**Figure 5:** A progressive reward comparison between EMT-CB and parametric, where each line corresponds to an average over 168 datasets in the OpenML repository. Reward is shown as a function of the number of environment interactions and vertical lines denote standard error. Parametric actually outperforms EMT on average, which we address with the hybridized algorithm discussed in Section 4.4.

**Figure 6:** A per-dataset comparison of final progressive reward between EMT-CB and parametric. Each point corresponds to a single dataset, and the vertical axis shows the difference between the two learners. The horizontal axis is ordered by this performance difference. Here we see that while there are many datasets for which parametric does better, there are 44 datasets where EMT is higher performing.

---

**Algorithm 4** Contextual Bandit EMT (EMT-CB)

---

**Initialize**: $c$ and $\epsilon$ as desired

**function** PREDICT($x_t \in \mathbb{R}^d$, $\mathcal{A}$)
    With probability $\epsilon$ **return** uniform random $a \in \mathcal{A}$
    Otherwise **return** $\arg\max_{a \in \mathcal{A}}$ EMT.Query($(x_t, a)$)
**end function**

**function** LEARN($x_t \in \mathbb{R}^d$, $a_t \in \mathcal{A}$, $y_t \in \mathbb{R}$)
    EMT.Learn($(x_t, a_t), y_t$)
**end function**

---

$z$, is the interaction of features in samples $x_1$ and $x_2$, instead of a dimension-wise difference. As a consequence, it is not the case that $\langle w, z \rangle$ is guaranteed to be zero for identical samples.

Figure 3 compares the performance of these algorithms on a dataset-by-dataset basis. The plotted points show the difference in total progressive reward averaged over random seeds. The plot shows that in 180 of the 206 datasets considered, self-consistency provides significant improvements. In the 6 where it does not, the performance difference is negligible.

Figure 4 compares the average progressive reward for these two algorithms over 168 datasets, with respect to the number of training examples. The number of datasets is slightly smaller here because we omit datasets with fewer than $4,000$ training examples. We can see that the self-consistent version has significantly better performance over all interactions on average.

## 4.2 COMPARISON TO ALTERNATIVE EPISODIC MEMORY DATA STRUCTURES

The most natural comparison to EMT is, as mentioned earlier, Contextual Memory Trees. The CMT implementation used throughout this work for CMT-CB is the official Vowpal Wabbit (VW) (Langford et al., 2007) implementation published by Sun et al. (2019). It is also worth noting that the original CMT paper evaluated the performance of CMT in an online framework with full-feedback, meaning that the agent receives information about the correct answer for each $x_t$ it encounters. In the partial-feedback scenarios studied here, where the agent only receives information corresponding to the executed action, we find CMT is far less performant overall.

Specifically, our results demonstrate that EMT's differences in routing and scoring endow it with performance superior to CMT in bandit scenarios. We see this both in terms of performance over time (Figure 5) and total performance on individual datasets (Table 1 and Appendix Figure 6) where EMT soundly outperforms CMT. This plot also compares to a parametric algorithm discussed in the proceeding section.

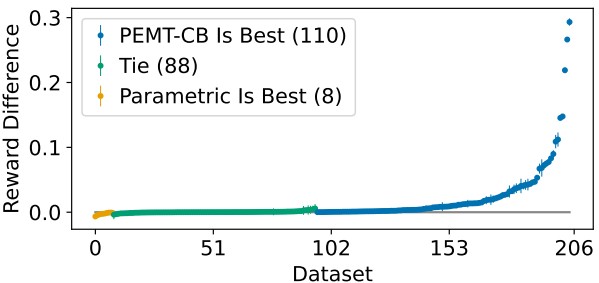

**Figure 7:** A per-dataset comparison of final progressive reward between PEMT-CB and parametric. Each point corresponds to a single dataset, and the vertical axis shows the difference between the two learners. The horizontal axis is ordered by this performance difference. PEMT-CB almost always performs at least as well as the parametric approach.

### 4.3 COMPARISON TO PARAMETRIC MODEL

Another natural comparison for this analysis is between EMT-CB and a parametric CB learner. A parametric approach could be thought of as encoding semantic memories, focusing on high-level concepts rather than specific experiences.

The parametric model is a linear regressor, taking context-action features as inputs and predicting expected rewards. The model is trained using SGD on the squared loss, as implemented by VW. Like EMT-CB and CMT-CB, this model is $\epsilon$-greedy, selecting uniform random actions with probability $\epsilon$ and selecting the action that yields the highest predicted reward with probability $1-\epsilon$. We use VW's default hyperparameter values (including $\epsilon = .1$) along with 1st and 2nd-order polynomial features. Table 1 and Figure 6 shows that EMT beats the parametric model on 44 cases (a minority).

The results here are reminiscent of the "no free lunch" theorem (Wolpert & Macready, 1997) which states there is no one universal learner that wins in all problems. Rather we must try and determine the most appropriate learner for each circumstance. The following subsection proposes a simple and high-performing method that is able to combine parametric and EMT-CB with almost no downsides.

To understand the dataset qualities that result in EMT-CB outperforming the parametric approach, we conducted a thorough meta-analysis, which is provided in Appendix 6. We find that the qualities most predictive of whether EMT will outperform parametric are (1) the mutual information between samples and labels and (2) how well the top eigenvector explains dataset variance. Intuitively, the former could be thought of as how reasonable it is to compare a new sample with the data stored at its corresponding leaf and the latter could be thought of as how reasonable it is for EMT to aggregate leaf samples based on the first principle component.

### 4.4 COMPARISON OF STACKED TO PARAMETRIC

Due to the mixed results of EMT-CB when compared against the parametric algorithm, we propose a simple method to combine the two learners, which we call PEMT-CB (and PCMT-CB when applying this method to CMT-CB). This combination is a simple stacking approach (Wolpert, 1992) where the rewards for each action predicted by EMT-CB are passed to the parametric CB model as a single additional feature. The parametric learner has no idea the additional feature came from EMT-CB and treats it simply as an additional parameter to optimize via SGD.

Figure 7 shows that PEMT-CB is able to outperform the parametric model on 110 individual datasets and under-performs by a very small margin in only 8 datasets. It is worth noting that stacking two learners in a partial feedback setting is more challenging than in a full-feedback setting because both learners may have different data needs for learning and not all actions can be explored. Previous work has sought to explicitly address this challenge from a statistical perspective (Agarwal et al., 2017), but here we find the simple stacking approach to be surprisingly powerful.

### 4.5 BOUNDED MEMORY ANALYSIS

For our final analysis we examine how sensitive EMT is to a memory budget, limiting the amount of memories EMT is permitted to keep. In order to maintain this limit we use a simple least-recently-used (LRU) strategy to eject a memory whenever our memory budget is exceeded. To get a sense of how the loss of memories might impact performance over time we filter our datasets down to the 116 which have $32,000$ or more training examples and evaluate out to $32,000$ iterations rather than the $4,000$ explored up to this point.

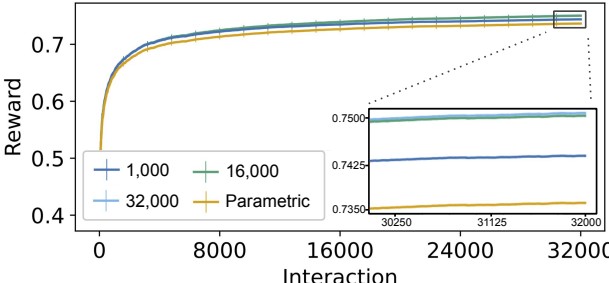

**Figure 8:** The performance loss between the unbounded PEMT-CB and bounded PEMT-CB algorithms is minimal even with a 3.1% memory budget. Additionally, at no budget level does PEMT-CB performance drop below the parametric model. Each line corresponds to an average over 116 datasets in the OpenML repository. Reward is a function of the number of environment inter-actions and vertical lines denote standard error.

We find that even with a budget of $1,000$ memories (with memory eviction via LRU) PEMT-CB still outperformed parametric alone with statistical significance on $61$ datasets (see Appendix Figure 14 for a per-dataset comparison). Figure 8 shows that a $16,000$-sample memory budget works almost exactly as well as the no-budget version of PEMT-CB. That is, equivalent performance can be attained despite the fact that the memory can only hold half as much information. Even when using an extremely conservative memory budget of $1,000$ samples, PEMT-CB only accrues a loss in reward of $.0075$ on average. Overall, very few memories are needed to consistently beat the parametric approach in isolation, and not much is needed to match the performance of the unbounded memory variant of PEMT-CB.

## 5 DISCUSSION AND FUTURE DIRECTIONS

**Summary** As discussed in Section 3, finding methods that effectively use episodic memory is an important and active area of research. This work proposes a new episodic memory model for sequential learning called Eigen Memory Trees (EMT). EMT possesses four distinguishing characteristics: (1) self-consistency, (2) incremental growth, (3) incremental learning, and (4) sub-linear computational complexity. EMT adds to an important but somewhat understudied area in online learning: efficient, online memory models.

To evaluate the effectiveness of EMT we converted $206$ datasets from OpenML into contextual bandit problems. Using these we compared EMT's performance to an alternative online memory model, CMT, and a parametric CB learner. EMT outperformed CMT across the board (see Figure 5) while outperforming parametric in $44$ of the $206$ (see Figure 6) datasets.

We further proposed PEMT-CB, a simple extension of EMT that is able to consistently outperform both EMT and parametric alone, even when forced to use a small memory budget (see Table 1). The aggregate strategy passes EMT reward predictions as a feature into the parametric learner. This result is important for a number of reasons. First, it is suggests that EMT offers a no-downside method for improving existing CB algorithms. Second, to our knowledge, this is the first work demonstrating that stacking can improve performance in the partial feedback setting of contextual bandits (cf. (Agarwal et al., 2017)).

Applied machine learning models often have to work with resources constrained by business and operational requirements. For memory models this is often the amount of memories that can be stored and efficiently accessed. We evaluated PEMT-CB performance for four memory budgets, showing that observed performance improvements of PEMT-CB are largely maintained even with an extremely constrained memory capacity (see Figure 8). Perhaps most importantly, constraining the memory budget had no negative impact on the performance of the parametric learner.

**Applications, limitations and future studies** The fast reactivity of memorization, when appropriate, is appealing in real world applications where data acquisition incurs natural costs via acting in the world. Thus, incorporating memorization in practice can broaden the applicability of contextual bandits, e.g., to scenarios in information retrieval and dialogue systems (Bouneffouf et al., 2020).

EMT also seems especially well suited for quick-changing environments. Rather than requiring multiple SGD steps to update, new memories are immediately accessible after insertion, and stale memories can be directly removed or added when situations change in known ways. This highlights both a limitation of this study as well as an opportunity: how robust is EMT to non-stationarity? The fixed nature of EMT's top-eigen routers along with our simple LRU eviction rule for memory budgeting likely make the model susceptible to performance degradation when environments shift.

## REPRODUCIBILITY STATEMENT

We have taken considerable measures to ensure results are as reproducible as possible. We chose to use a large number of publicly available datasets with many replications. We have also provided our algorithm code, experiment code, result data, and plotting code in the supplement. The experiment code we provided, when executed, will download all appropriate datasets, apply all data transformations described in the paper, and create 50 shuffled replicates for each data set using the same random seeds used by the authors. Finally, our supplementary code also includes a conda environment file to help future researchers recreate our development environment on their machines when running the experiments.

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

# Appendix

## 6 META-DATA ANALYSIS

An analysis was performed to understand the characteristics of the datasets where EMT-CB outperformed the parametric CB learner. Twelve dataset features were selected from the meta-learning literature. These twelve features are described in Table 2.

We trained a random forest (RF) model using the features in Table 2 to predict when EMT would outperform the parametric learner. This model was able to predict winning CB model with a macro

**Table 2:** The features used to analyze why EMT outperformed the parametric learner on some datasets.

| Feature | Reference | Description |
|---|---|---|
| Top Eigenvector Explained | | The percentage of dataset variance explained by the top eigenvector. |
| Eigenvector Count To Explain 95% Var | Lorena et al. (2019) | The number of eigenvectors required to explain 95% of the feature variance. |
| Mean Mutual Information Feature/Label | Castiello et al. (2005); Reif et al. (2014) | The average amount of mutual information between each feature and the target label. |
| 1 Nearest Neighbor Accuracy | Reif et al. (2014); Abdelmessih et al. (2010) | The accuracy of a 1 nearest neighbors classifier on the dataset. |
| Single Node Decision Tree Accuracy | Reif et al. (2014); Abdelmessih et al. (2010) | The accuracy of a single node decision tree on the dataset. |
| Naive Bayes Accuracy | Reif et al. (2014); Abdelmessih et al. (2010) | The accuracy of a naive Bayes classifier on the dataset. |
| Normed Label Entropy | Lorena et al. (2019); Castiello et al. (2005); Reif et al. (2014) | The amount of entropy in the class distribution (normed so that 1 is maximum entropy). |
| Noise to Signal Ratio | Castiello et al. (2005) | Ratio extra feature entropy to mutual information between features and labels. |
| Binary Feature Count | | The count of the number of binary features in the dataset. For one hot encoded categorical features this counts every level. |
| Percent of Nonzero Features | Lorena et al. (2019) | The percentage of non-zero features in the dataset. |
| Efficiency of Best Single Feature | Lorena et al. (2019) | A measure of the single best feature's ability to distinguish classes. The efficiency is averaged across all one-v-one label matchups. |
| Maximum Fisher's Discriminant Ratio | Lorena et al. (2019) | A Measure of feature overlap when discriminating class labels. The ratio is averaged across all one-v-one label match-ups. |

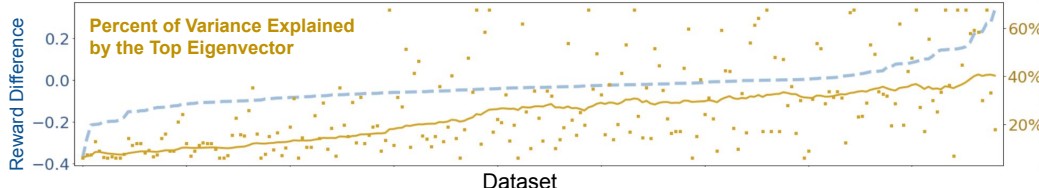

**Figure 9:** This plot visualizes the relationship between the variance explained by the top-eigenvector for a dataset and the performance of EMT-CB versus parametric. The x-axis represents our study datasets. The blue line shows the difference in performance between EMT-CB and parametric. The yellow points show the feature values for each dataset and the yellow line show the rolling average of all the yellow points.

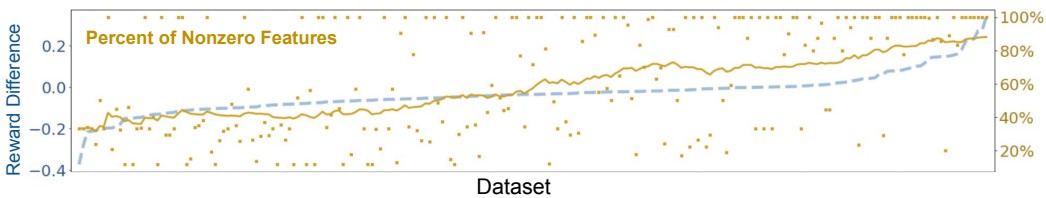

**Figure 10:** This plot visualizes the relationship between the percent of nonzero features in a dataset and and the performance of EMT-CB versus parametric. The x-axis represents our study datasets. The blue line shows the difference in performance between EMT-CB and parametric. The yellow points show the feature values for each dataset and the yellow line show the rolling average of all the yellow points.

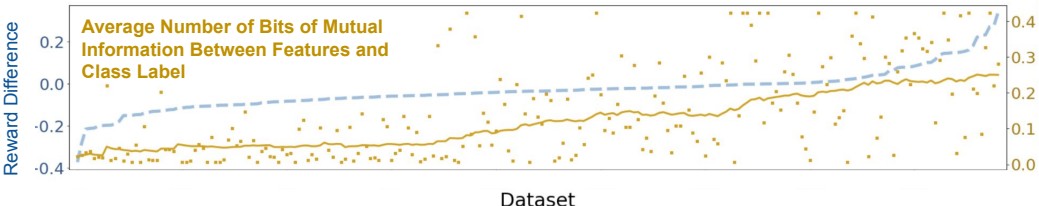

**Figure 11:** This plot visualizes the relationship between the mutual information metadata feature and the performance of EMT-CB vs parametric. The x-axis represents our study datasets. The blue line shows the difference in performance between EMT-CB and parametric. The yellow points show the feature values for each dataset and the yellow line show the rolling average of all the yellow points.

f1 score of .82 on held-out datasets. Figure 12 shows the average reduction in Gini impurity for each feature.

We then trained a compact RF model to see the minimum features needed. We used repeated k-fold cross validation with feature selection applied to the training split before fitting our RF model. The final reduced model had an f1 score of .79 on held out data using only 3 features: percent of variance explained by the top eigenvector, percent of nonzero features, average amount of mutual information between features and labels (measured in bits). We now look at these more closely.

The first of these features measures the explanatory power of the top eigenvector. We can see that this has a positive correlation with the performance of memory in Figure 9. We believe these datasets with a more informative top eigenvector for the whole dataset likely have more informative top eigenvector routers leading to improved tree searches for the EMT.

The second of these features is the percentage of non-zero values in a dataset. For this feature we observed that the fewer non-zero features in a dataset the better EMT performed (see Figure 10). When we looked at datasets with a high percentage of zero values the most common cause were large amounts of one hot encoded categorical features. Categorical variables with a large number of levels are naturally difficult to compare via a metric since each level is equidistant from every other level. In this case learning an effective scorer would become very challenging parametric would have a greater chance of beating EMT-CB.

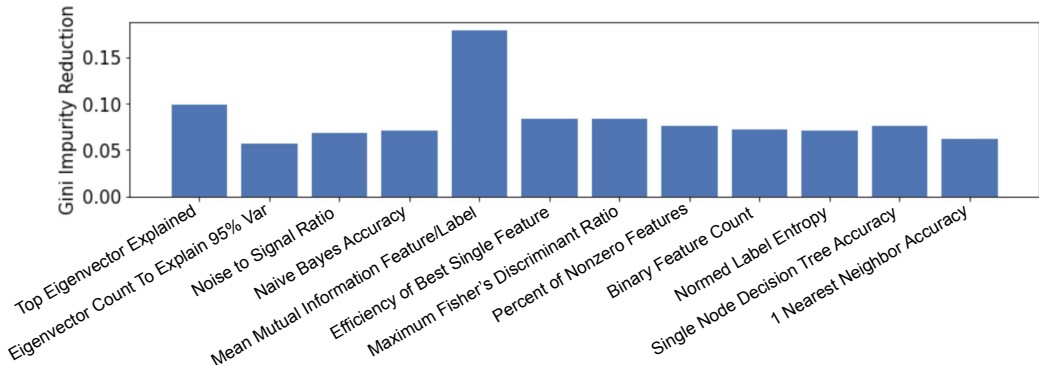

**Figure 12:** The average decrease in impurity in an RF model predicting whether EMT outperforms parametric.

The final feature in our compact model was the mutual information between individual features and a dataset's class (see Figure 11). This is a non-normalized value so datasets with more labels and fewer non-zero features will likely have higher values of this feature. For example, a dataset with a binary label has an upper limit of $1$ for this feature while a dataset with $30$ class labels has an upper limit of $4.91$. High values in this variable also indicate that there is a strong statistical relationship between features and class labels but place no restriction on the functional that relationship may take.

# 7 SUPPLEMENTARY FIGURES

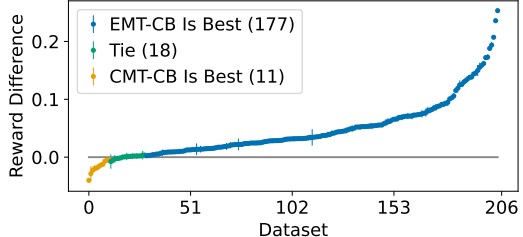

**Figure 13:** A per-dataset comparison of final progressive reward. Each point corresponds to a single dataset. The y-axis shows the difference between the two learners. The x-axis is ordered by this performance difference.

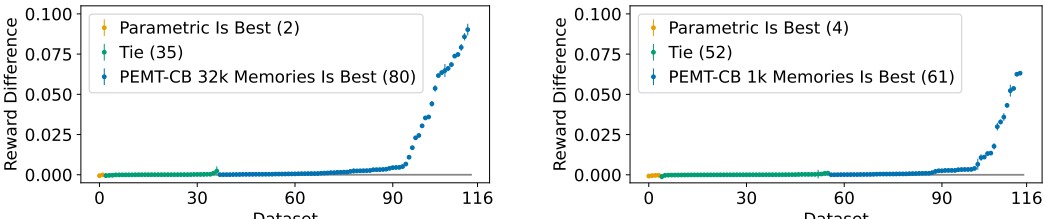

**Figure 14:** A per-dataset comparison of final progressive reward between memory budgeted PEMT-CB and parametric. The left plot shows this difference when there is no bounding. The right plot shows the difference when bounding to 1k memories. Each point corresponds to a single dataset. The y-axis shows the difference between the two learners. The x-axis is ordered by this performance difference.

