# OpenReview forum: "Eigen Memory Trees"
_ICLR.cc/2023/Conference — Submitted to ICLR 2023_

### Official Review · Reviewer_UauN · 2022-10-29

**Confidence:** 3
**Correctness:** 4
**Technical Novelty And Significance:** 1
**Empirical Novelty And Significance:** 2
**Recommendation:** 3

**Clarity, Quality, Novelty And Reproducibility:**

The paper is well written but could improve its visualization. Some visualization of the data structure, such as showing how data is partitioned by nodes (using some low-dimensional projection like t-SNE for example) could help the reader understand the inner working of the approach.

**Strength And Weaknesses:**

Strengths
- The analysis of memory properties is well executed and the algorithm motivation is clear
- Code is provided for full reproducibility

Weaknesses
- Method is very close to CMT
- The comparison is kind of limited given that modern libraries such as FAISS could be used for example

**Summary Of The Paper:**

This work introduces a new data structure based on a binary tree. Each node is split using PCA on the node data. With respect to existing data structure EMT are easier to learn in streaming tasks. The most similar work to EMT are CMT which have similar structure internally and pretty identical mechanism to route and score data. EMT differ in term of self-consistency guaranteeing such property.

Experiments are performed on bandits as a use case for streaming tasks requiring a memory structure.

EMT alone is not really competitive with respect to other mechanism on bandit problems, but the parametric variant is able to outperform many existing memory structures on most of the 206 tasks taken into consideration.



**Summary Of The Review:**

The paper is clearly explained and the method has some interesting properties, unfortunately to outperform the competition the structure must be stacked with an existing parametric approach. Moreover the approach is also very close to CMT. Some other data structures, e.g. the plethora of indices in FAISS could be taken into consideration when performing the experiments.

---

> ### Author Response · Authors · 2022-11-15
> **Response to Reviewer UauN**
>
> Thank you for taking the time to review. We believe the paper has been misunderstood, unfortunately.
>
> 1. The routing mechanism in EMT (median of the top eigenvector) is quite different from the routing mechanism in CMT (online learning to create a balanced split).  As we demonstrate (Section 4.2), this difference is tremendously impactful in terms of empirical performance.
>
> 2. The unsupervised algorithms implemented in FAISS solve a different problem than what is being addressed in this work for the partial-feedback setting. Every algorithm in FAISS requires a user-specified similarity metric while EMT crucially does not.  In our view, whether or not it is possible to create a sensible and effective indexing system based on learning and generalization rather than a metric structure is a critical question of interest in neuroscience and machine learning more broadly. This paper is concerned with effectively learning these indices via partial feedback.
>
> 3. Empirically, the parametric learner is not a competing approach to the EMT. Rather, we view the parametric comparisons as providing insight into when memory is or is not useful for solving problems. We see the competing model as CMT (i.e., an online incremental memory model) which EMT beats both directly and in stacked experiments.
>
> 4. We agree that a t-SNE visualization of the data structure is elucidating. We have updated our paper to include Figure 2, showing how data are distributed across EMT for OpenML dataset BNG(breast-w), to address this point.

---

### Official Review · Reviewer_RGJi · 2022-10-30

**Confidence:** 2
**Clarity, Quality, Novelty And Reproducibility:** The paper is well-written and easy to…
**Correctness:** 3
**Technical Novelty And Significance:** 2
**Empirical Novelty And Significance:** 3
**Recommendation:** 6

**Strength And Weaknesses:**

Overall, the paper is well-written and easy to follow. The experimental results demonstrate that the proposed method is promising. The authors use fixed top-eigenvector routers, pairwise feature differences, and a loss function according to ranking in EMT, enabling CMT improvement.

In Section 4.5, even with an extremely conservative memory budget of 1,000 samples, PEMTCB only incurs a reward loss of 0.0075 on average. Very little memory can beat parametric methods in isolation. Is this conclusion related to the data set? If you can try the performance on these data sets~ (AmazonCat, Wiki10, Amazon [1]), the conclusion will be more convincing.

A typo:
1. “eThe EMT incorporates two particular design decisions”

[1]. Jasinska-Kobus, Kalina, et al. "Online probabilistic label trees." International Conference on Artificial Intelligence and Statistics. PMLR, 2021.

**Summary Of The Paper:**

This work proposes a novel episodic memory model for sequential learning, termed an Eigen Memory Tree (EMT). EMT adds to an important but under-studied topic in online learning: efficient online memory models. To evaluate the effectiveness of EMT, the authors tested 206 datasets from OpenML and compared with another online memory model, CMT, and a parametric CB learner. EMT outperforms CMT across the board, while outperforming parameters in 44 out of 206 datasets.

**Summary Of The Review:**

See Strength & Weaknesses

---

> ### Author Response · Authors · 2022-11-15
> **Response to Reviewer RGJi**
>
> Thank you for the review.
>
> 1. Unfortunately, the large number of labels in these datasets prevents partial-feedback approaches from being effective. For comparison, the CMT paper (along with the paper you cite) only evaluates in a full-feedback setting. If there are any other salient datasets in the literature that we’ve missed, ideally with a smaller number of total labels, we would be happy to include them.
>
> 2. Regarding the results with bounded memory: Yes, the plots we show average across all 206 of our datasets and individual performance fluctuates somewhat by dataset. The per-dataset plot comparing bounded PEMT-CB to parametric can be found in the appendix with Figure 14 if you’re interested. We also improved the caption on Figure 14 to make it clear it addresses your question. We reference Figure 14 in the text on page 9.
>
> 3. Thanks for catching the typo. We’ve fixed it in a resubmitted document.

---

### Official Review · Reviewer_ibPD · 2022-11-10

**Confidence:** 2
**Correctness:** 3
**Technical Novelty And Significance:** 2
**Empirical Novelty And Significance:** 2
**Recommendation:** 3

**Clarity, Quality, Novelty And Reproducibility:**

The writing is fine. Similar data structures like K-D tree and trees using projection exist. I think this should be evaluated in such context which should be the focus of a different community.


**Strength And Weaknesses:**

Pros: potentially useful.

Cons: I am not convinced ICLR is the right place for this paper. While there is connection to machine learning, the key contribution of a more efficient and practical data structure should be evaluated by community who focus on such issues. I believe database would be a better venue. It is hard to evaluate such work in the context of ICLR without comparing to other similar data structures.



**Summary Of The Paper:**

The paper proposes Eigen Memory Tree to serve as an efficient data structure for online memory.

**Summary Of The Review:**

I don't think ICLR is the right venue for this paper.

---

> ### Author Response · Authors · 2022-11-14
> **Response to Reviewer ibPD**
>
> First, thank you for your time. Unfortunately, we feel the paper has been misunderstood.
>
> 1. EMT is best understood as a memory module similar to the work in [1] rather than a K-D tree or projection tree. EMT creates an index based on learning and generalization (as in [1]) rather than using a pre-defined metric, and thus solves a different problem than these data structures. Please see point 2 in our response to Reviewer UauN.
>
> 2. Separately, we strongly disagree that a machine learning conference is not the right venue for work of this sort. The most natural comparison to our algorithm, CMT, was published in ICML 2019. Our related work section overviews many relevant approaches in the space of contextual bandits and learned memory systems — all of these have been published in machine learning venues. EMT is about efficiently learning in partial feedback scenarios and belongs in a conference where these topics are discussed.
>
> [1] Zhu, Guangxiang, et al. "Episodic Reinforcement Learning with Associative Memory." International Conference on Learning Representations. 2019.

---

### Author Response · Authors · 2022-11-14
**To All Reviewers**

Thank you for your time and feedback. We respond to your individual comments below.

Also, our resubmitted document fixes some typos, adds a suggested figure and improves the caption on an existing figure.

---

### Decision · Program_Chairs · 2023-01-20

**Decision:**

Reject

**Justification For Why Not Higher Score:**

The messages should have been spelled out more clearly. It is left to the reader to decipher what is the contributions to be remembered from the reading (it is a shame because there seems to be potential here).

**Justification For Why Not Lower Score:**

Contribution touching to online learning are key, nowadays, and having appropriate structures to help work in these settings is important.

**Metareview: Summary, Strengths And Weaknesses:**

This introduces a new model Eigen Memory Tree that is a tree-structure capable of doing online memorizing of data at low cost, and that can be used in sequential learning scenarios. The model is used in a series of contextual bandit expriments and shows how it can help have more efficient and effective learning in those situations.

Strengths
- providing devoted ML models/structures that help cope with online scenario is pivotal for the ML community
- empirical results support the relevance of the structure

Weaknesses
- the authors did not take the appropriate time to state the problem tackled here: do they look for a data structure to store data? do they look for a data structure to do online learning (i.e. a data structure with predictive power)?
- the write-up is very ascending: all the low-level procedures are provided, they are commented and described but it is hard to understand how they work together and to what end they are assembled. It might have been more enlightening for the reader to understand first the end-goal and a high level procedure (like a "main") and then crawl down to the more technical procedures
- there is a lack of formal analysis/statement that show/establish the low computational complexity of using the proposed structure

All in all, the main drawback of the paper is for the authors to have not been clearer in the contribution as to the main take-home messages. This is a central reason that may explain the feedbacks of the reviewers, that were not capable of getting a hold on a main message.